# Cathepsin B aggravates atherosclerosis in ApoE-deficient mice by modulating vascular smooth muscle cell pyroptosis through NF-κB / NLRP3 signaling pathway

Hui Li[1,2,3], Quanwei Zhao[1,2,3], Danan Liu[1,2]*, Bo Zhou[1,2,3], Fujun Liao[1,2,3], Long Chen[1,2,3]

1 Department of Cardiology, Affiliated Hospital of Guizhou Medical University, Guiyang, Guizhou, China,
2 Institute of Medical Sciences, Guizhou Medical University, Guiyang, Guizhou, China, 3 School of Clinical Medicine, Guizhou Medical University, Guiyang, Guizhou, China

* liudanan2000@163.com

## Abstract

Atherosclerosis (AS) is a chronic inflammatory disease involving cell death and inflammatory responses. Pyroptosis, a newly discovered pro-inflammatory programmed cell death process, exacerbates inflammatory responses. However, the roles of cathepsin B (CTSB) in pyroptosis and AS remain unclear. To gain further insight, we fed ApoE$^{-/-}$ mice a high-fat diet to investigate the effects and mechanisms of CTSB overexpression and silencing on AS. We also explored the specific role of CTSB in vascular smooth muscle cells (VSMCs) *in vitro*. The study revealed that high-fat diet led to the formation of AS plaques, and CTSB was found to increase the AS plaque lesion area. Immunohistochemical and TUNEL/caspase-1 staining revealed the existence of pyroptosis in atherosclerotic plaques, particularly in VSMCs. *In vitro* studies, including Hoechst 33342/propidium iodide staining, a lactate dehydrogenase (LDH) release assay, detection of protein indicators of pyroptosis, and detection of interleukin-1β (IL-1β) in cell culture medium, demonstrated that oxidized low-density lipoprotein (ox-LDL) induced VSMC pyroptosis. Additionally, CTSB promoted VSMC pyroptosis. Ox-LDL increased the expression of CTSB, which in turn activated the NOD-like receptor protein 3 (NLRP3) inflammasome and promoted NLRP3 expression by facilitating nuclear factor kappa B (NF-κB) p65 nuclear translocation. This effect could be attenuated by the NF-κB inhibitor SN50. Our research found that CTSB not only promotes VSMC pyroptosis by activating the NLRP3 inflammasome, but also increases the expression of NLRP3.

## 1. Introduction

Atherosclerosis (AS) is a chronic inflammatory disease that occurs in large- and medium-sized arteries [1]. It is characterized by the accumulation of lipids in the arterial wall and the formation of atherosclerotic plaques that obstruct the vascular lumen, leading to tissue and organ

**Data Availability Statement:** All relevant data are within the paper and its Supporting Information files.

**Funding:** Danan Liu plays a role in study design, decision to publish and preparation of the manuscript. Danan Liu provided funding support. This study was supported by the National Natural Science Foundation of China (81660083), Guizhou Science and Technology Innovation Talent Team Project [Guizhou Science and Technology Cooperation Platform Talent (2020)5014]; Guizhou Province "Hundred" level innovative talents Training Program project [Guizhou Science and Technology Cooperation Talents (2015) No. 4026]. The funders had no role in the study design, data collection and analysis, decision to publish, or preparation of the manuscript.

**Competing interests:** The authors have declared that no competing interests exist.

hypoxia [2]. AS is the common pathological basis of coronary heart disease, stroke, peripheral vascular occlusion, and other atherosclerotic vascular diseases (ASCVDs) [3, 4]. The occurrence and development of AS is accompanied by cell death and inflammation.

Pyroptosis, a newly discovered pro-inflammatory programmed cell death process, is divided into two pathways: the canonical inflammasome pathway, which is dependent on caspase-1, and the non-canonical inflammasome pathway, which is dependent on caspase-4/5/11 [5]. The NOD-like receptor protein 3 (NLRP3) inflammasome belongs to the nucleotide-binding oligomerization domain-like receptor (NLR) family and is a crucial component of the canonical inflammasome pathway [6]. Activation of NLRP3 inflammasomes results in the conversion of pro-caspase-1 to cleaved-caspase-1. The role of cleaved-caspase-1 is twofold. First, it facilitates the maturation of inflammatory mediators such as interleukin (IL)-18 and IL-1β. Second, it promotes the cleavage of gasdermin D (GSDMD) to form N-terminal gasdermin D (GSDMD-N), which in turn disrupts the integrity of the cell membrane and leads to the release of inflammatory mediators [5, 7]. Pyroptosis-induced cell death not only results in the loss of physiological function but also triggers the production of inflammatory mediators that can further exacerbate the inflammatory response. Within the vascular wall and atherosclerotic lesions, vascular smooth muscle cells (VSMCs) are particularly crucial. In addition to maintaining the normal structure of vascular walls, VSMCs play a role in the development of AS through processes such as proliferation, migration, phenotypic transformation, and secretion of pro-inflammatory and pro-proliferative factors. These actions contribute to endothelial cell injury, promote foam cell formation, and facilitate the formation of fiber caps [8–10]. VSMCs and the extracellular matrix (ECM) that they secrete constitute the protective fibrous cap in AS plaques. Death of VSMCs and breakdown of the ECM due to inflammation are the primary factors that contribute to plaque instability [11, 12]. Consequently, the condition of the VSMCs is crucial for the onset and progression of AS.

The cathepsins are a family of proteolytic enzymes with rich functions in lysosomes and are classified based on the amino acid residues present in their active sites, including metallo, serine, threonine, aspartic, and cysteine proteases [13]. Cathepsin B (CTSB) is a cysteine protease widely expressed in human tissues and cells, and has diverse biological roles in the lysosomes, cytoplasm, and ECM [14]. Various studies have investigated the cathepsin activity in AS, and cathepsins have been suggested to play a causal role in the development of AS. Kim et al. [15] and Abd-Elrahman et al. [16] discovered that CTSB is highly expressed in human carotid plaques and stroma, and its expression and activity are correlated with the severity of plaques and symptomatic events. Wuopio et al. [17] conducted an analysis of the CLARICOR trial substudy and found that CTSB was associated with an elevated risk of cardiovascular events in patients with stable coronary heart disease. Mareti et al. [18] revealed that the expression of CTSB in human plasma is associated with AS and ASCVD and that CTSB is involved in the early mechanisms of atherogenesis and continuum of the atherosclerotic process, potentially leading to the development of clinical events. Lysosomal damage is a significant contributor to NLRP3 inflammasome activation, as supported by numerous studies indicating that lysosomal cysteine CTSB can facilitate this process [19–21]. However, it remains unclear whether CTSB exacerbates AS development by promoting NLRP3 inflammasome activation and aggravating VSMC pyroptosis.

The nuclear factor kappa B (NF-κB) pathway is a classical inflammatory signaling pathway. In its inactive state, NF-κB exists in the cytoplasm as a p50/p65 dimer, which is inhibited by interaction with inhibitor of κB (IκB). Upon extracellular stimulation, NF-κB-p65 is phosphorylated and dissociates from IκB, forming p-NF-κB-p65. The translocation of NF-κB-p65 to the nucleus promotes the transcription of various inflammatory genes including tumor necrosis factor-alpha (TNF-α), interferon-gamma (IFN-γ), IL-6, NLRP3, and IL-1β [22].

However, whether CTSB promotes NLRP3 expression by activating NF-κB has not yet been confirmed.

In this study, lentivirus vectors were used to overexpress and silence CTSB in ApoE-/- mice and VSMCs. To investigate the role and potential mechanism of CTSB in NLRP3-mediated VSMC pyroptosis, the NF-κB-p65 nuclear translocation inhibitor SN50 was used to inhibit NF-κB activation in VSMCs. The findings of this study provide a theoretical basis for further understanding the pathogenesis of AS and exploring new targets for AS treatment.

## 2. Materials and method

### 2.1 Materials

CTSB overexpression and silencing lentivirus vectors (Lv-CTSB and sh-CTSB, respectively) and the empty lentiviral vector were obtained from OBIO Technology (Shanghai, China). SN50 (HY-P0151) was purchased from MedChem Express (Monmouth Junction, NJ, USA). Oxidized low-density lipoprotein (ox-LDL, YB-002) was purchased from Yiyuan Biotechnology (Guangzhou, China). The hematoxylin and eosin (HE) stain kit (G1120), Masson's Trichrome Stain Kit (G1340), RIPA buffer (R0010), 1% (v/v) penicillin-streptomycin mixture (P1400), Hoechst 33342/PI Double Stain Kit (CA1120), 4′,6-diamidino-2-phenylindole (DAPI, S2110), and IL-1β enzyme-linked immunosorbent assay (ELISA) Kit (SEKM-0002) were purchased from Solarbio Science & Technology (Beijing, China). The lactate dehydrogenase (LDH) Activity Assay Kit (C0016) and puromycin (ST551-10 mg) were purchased from Beyotime Biotechnology (Shanghai, China). Primary antibodies against CTSB (31718), caspase-1 (83383), apoptosis-associated speck-like protein containing a C-terminal caspase recruitment domain (ASC, 67824), GSDMD-N (10137), and IL-18 (57058) were purchased from Cell Signaling Technology (Danvers, MA, USA). Antibodies against NF-κB-p65 (ab32536), p-NF-κB-p65 (ab76302), IκB-α (ab32518), p-IκB-α (ab133462), glyceraldehyde 3-phosphate dehydrogenase (GAPDH, ab181602), goat anti-rabbit IgG H&L(HRP) (ab6721), goat anti-rabbit IgG H&L(FITC) (ab6717), and goat anti-rabbit IgG H&L(Alexa Fluor 594) (ab150080) were obtained from Abcam (Cambridge, UK). Antibodies against NLRP3 (YT5382), GSDMD-N (YT7991), and IL-1β (YT5201) were obtained from ImmunoWay Biotechnology Company (Plano, TX, USA). Antibodies against caspase-1 p10 (AF4022) and alpha-smooth muscle actin (α-SMA, AF1032) were obtained from Affinity Biosciences (Cincinnati, OH, USA). The reverse transcription kit and SYBR Premix Ex Taq™ kit were obtained from TaKaRa (Beijing, China). Primer synthesis and sequencing were conducted by Nanjing Tsingke Biotechnology (Nanjing, China). An autofluorescence quenching kit (SP-8400) was purchased from Vector Laboratories (Newark, CA, USA).

### 2.2 Mice

Male ApoE$^{-/-}$ mice (6 weeks old, n = 60) were purchased from SPF Biotechnology (Beijing, China). The mice were randomly divided into five groups (n = 12 per group). (1) Negative control group (NC), (2) high-fat diet group (HF), (3) mock group (HF+GFP-tagged neutral construct), (4) lentivirus CTSB group (HF+Lv-CTSB), and (5) sh-CTSB group (HF+sh-CTSB). The mice were housed in the SPF Animal Laboratory of Guizhou Medical University at 22–24°C, 55–60% humidity, and 12 h of light/dark. The mice were given free access to food and water and were fed with chow diet for 2 weeks. Then, the negative control group was fed the chow diet, while all other groups were fed a 16% fat and 1.3% cholesterol (high-fat) diet for 12 weeks. At 8 weeks of age, the mice were injected via tail vein with empty (GFP-tagged neutral construct), CTSB-overexpressing, or CTSB-silencing lentivirus vector at a dose of $2 \times 10^7$ TU per mouse. Two weeks after viral transduction, orbital blood was collected under pentobarbital

sodium anesthesia. A subset of mice was then euthanized by cervical dislocation. The heart and aorta were isolated to evaluate the virus transduction efficiency. The remaining mice were euthanized at 20 weeks of age using the same method. This study was carried out in strict accordance with the recommendations of the Guide for the Care and Use of Laboratory Animals of the National Institutes of Health. The study protocol was approved by the Committee on the Ethics of Animal Experiments of Guizhou Medical University (Protocol Number:2200642). All surgeries were performed under sodium pentobarbital anesthesia, and efforts were made to minimize suffering.

## 2.3 Blood lipid test

Blood samples were obtained from the orbital veins of the mice, and serum was analyzed using low-density lipoprotein (LDL) cholesterol kits according to the manufacturer's instructions.

## 2.4 Oil Red O staining

The isolated mouse heart and aorta were excised from the aortic root and were either fixed with 4% paraformaldehyde for paraffin sections or frozen at -80˚C for frozen sections. The aortas were stained with Oil Red O. At least three aortas per group were used to measure the Oil Red O-positive area and total area of the luminal surface. Image J software was used for the analysis.

## 2.5 Fluorescence detection after lentivirus transduction

The frozen sections were warmed to room temperature in the dark. After washing with phosphate-buffered saline (PBS), DAPI staining solution was added, and the cells were incubated at room temperature in the dark for 10 min. After washing away the DAPI dye solution with PBS, images were observed under a fluorescence microscope.

## 2.6 HE and Masson's staining

Five micrometer-thick paraffin sections were mounted on slides and placed in xylene and gradient alcohol for dewaxing and rehydration, respectively. The sections were then stained with the HE stain kit and Masson's Trichrome Stain Kit, following the manufacturer's instructions.

## 2.7 Immunohistochemistry staining

Paraffin sections were dewaxed and rehydrated according to the method described in section 2.6. The sections were then heated to 94˚C for 20 min in 0.1 M citric acid antigen retrieval solution and cooled to room temperature. The sections were subsequently incubated with 3% hydrogen peroxide at room temperature in the dark for 25 min to block endogenous peroxidase, blocked with 3% bovine serum albumin (BSA) at room temperature for 30 min, and then incubated overnight at 4˚C with primary antibody (CTSB antibody, 1:200; NLRP3 antibody, 1:200; caspase-1 p10 antibody, 1:200; GSDMD-N antibody, 1:200). After washing, the sections were incubated with secondary antibody (goat anti-rabbit IgG H&L(HRP) antibody, 1:1000) at room temperature for 90 min. After washing, 3,3′-diaminobenzidine (DAB) and hematoxylin staining were performed, and the sections were photographed under a microscope. ImageJ software was used for analysis.

## 2.8 TUNEL and caspase-1 p10 immunofluorescence double staining

The paraffin-embedded sections were deparaffinized and rehydrated as described in section 2.6. The sections were incubated with proteinase K working solution at 37˚C for 30 min and

permeabilized with 0.1% Triton X-100 at room temperature for 20 min. Terminal deoxynucleotidyl transferase, dUTP, and buffer were mixed at a ratio of 1:5:50 and incubated with the sections at 37˚C for 2 h. The sections were then blocked with 3% BSA for 30 min and incubated with caspase-1 p10 (1:200) primary antibody overnight at 4˚C. After washing, an Alexa Fluor594-conjugated secondary antibody (1:500) was added and incubated at room temperature for 2 h. An autofluorescence quenching kit (SP-8400, Vector Laboratories) was used to quench tissue autofluorescence according to the manufacturer's instructions, and the nuclei were stained with DAPI. The sections were photographed under a fluorescence microscope.

## 2.9 Immunofluorescence staining

Immunofluorescence staining was performed on tissue cross-sections treated with different intervention factors. The slide-mounted sections were fixed with 4% paraformaldehyde for 15 min, incubated with 0.5% Triton X-100 for 20 min for cell permeabilization, blocked with 3% BSA for 30 min, incubated with NF-κB-p65 primary antibody (1:100) overnight at 4˚C, and incubated with Alexa Fluor 594-conjugated secondary antibody (1:500) at room temperature for 2 h. DAPI was used to stain nuclei after adding tissue autofluorescence quencher. Images were collected using a fluorescence microscope or a laser scanning confocal microscope.

## 2.10 Cell culture and transduction

The MOVAS cell line (VSMCs derived from murine aortic vascular smooth muscle; Shanghai Fuheng Biotechnology Co., Ltd., Shanghai, China) from passages 5–15 was used for all experiments. The VSMCs were cultured in complete DMEM (Gibco, 11965092) containing 10% fetal bovine serum (Gibco, A3161002C) and 1% (v/v) penicillin-streptomycin mixture in a 95% air / 5% $CO_2$ incubator at 37˚C. For the transduction experiments, the VSMCs were inoculated into a 24-well plate at $5\times10^4$ cells/well. Lv-CTSB, sh-CTSB and empty lentiviral vectors were used to transduce the cells according to the manufacturer's protocol. After 12 h, the medium was replaced with fresh complete medium, and puromycin (1 μg/ml final) was added for selection. After 6 days, the mRNA and protein expression of CTSB in the transduced cells was assessed. Stably transduced cells were selected for subsequent experiments in which the VSMCs were treated for 12 h with SN50 (75 μg/ml) and for 24 h with ox-LDL (150 mg/ml).

## 2.11 Reverse transcription-quantitative polymerase chain reaction (RT-qPCR)

Total RNA was extracted from cells using TRIzol and reverse transcribed into cDNA qPCR was then performed using the SYBR Premix Ex Taq™ kit. The reaction conditions were as follows: pre-denaturation at 95˚C for 1 min, 40 cycles of denaturation at 95˚C for 15 s and annealing at 60˚C for 1 min. Nonspecific amplification was excluded by melting and amplification curve analyses, and the reaction was repeated three times for each sample. With *GAPDH* as an internal reference, the relative expression level of the target gene was calculated using the $2^{-\triangle\triangle Ct}$ method. The primer sequences were as follows: *GAPDH* F:5′–TTCACCACCATGGAGA AGGC–3′, R:5′–TGAAGTCGCAGGAGACAACC–3′; target gene *CTSB* F:5′–TCCTTGATCCTTC TTTCTTGCC–3′, R:5′–ACAGTGCCACACAGCTTCTTC–3′.

## 2.12 Cell death assay

For Hoechst 33342/propidium iodide (PI) staining, after the designated treatments, the VSMCs were incubated with a mixture of Hoechst 33342 and PI for 20 min in an ice bath and photographed using a fluorescence microscope.

## 2.13 Western blotting

Total protein was extracted from cells and tissues using RIPA lysis buffer and 1% (v/v) protease inhibitor cocktail. After boiling in sodium dodecyl sulfate (SDS) loading buffer, the same volume of samples was separated by 10% or 12% sodium dodecyl sulfate-polyacrylamide gel electrophoresis (SDS-PAGE). The separated proteins were transferred onto a polyvinylidene fluoride (PVDF) membrane (Millipore, Billerica, MA, USA), which was blocked with 5% skim milk for 1 h at room temperature, washed with Tris-buffered saline-tween (TBST) for 30 min, and incubated with primary antibodies overnight at 4°C. After washing with TBST for 30 min, the membranes were incubated with secondary antibodies for 1 h at room temperature. Finally, the protein bands were detected using an ECL system and analyzed using ImageJ software.

## 2.14 LDH release assay

LDH released into cell culture medium was measured using an LDH Assay Kit according to the manufacturer's instructions.

## 2.15 ELISA

The levels of IL-1β in mouse serum and cell culture medium were measured using an ELISA kit according to the manufacturer's instructions.

## 2.16 Statistical analysis

Each experiment was independently repeated at least three times. GraphPad Prism 9.0 (GraphPad Software, La Jolla, CA, USA) was used for statistical analyses. The experimental data are presented as means ± standard error ($\chi \pm$ SEM) and are compared among groups using one-way analysis of variance (ANOVA). $P<0.05$ was considered statistically significant.

# 3. Results

## 3.1 Lentiviral transduction of ApoE$^{-/-}$ mice and VSMCs

Two weeks and 20 weeks after lentiviral transduction of ApoE$^{-/-}$ mice, we detected the expression of the mouse aortic construct. The vascular wall exhibited EGFP fluorescence in frozen sections of the aortic root (Fig 1A). In addition, all stably transduced VSMC lines that were puromycin resistant also exhibited EGFP fluorescence (Fig 1B). The results of RT-qPCR and western blot analyses showed a significant difference in the expression of *CTSB* mRNA and protein levels in the mouse aorta (Fig 1C, 1E and 1G) and VSMCs (Fig 1D, 1F and 1H) when compared to the mock group. These results indicate that Lv-CTSB and sh-CTSB transduction was successful in the ApoE$^{-/-}$ mouse aortic wall and VSMCs. We also detected the expression of α-SMA in VSMCs between the 5th and 15th passage by immunofluorescence. The results showed that there was no significant difference in the expression of α-SMA among these cell passages (Fig 1I and 1J).

## 3.2 High-fat diet promotes plaque formation and increases LDL levels in ApoE$^{-/-}$ mice

After 12 weeks of high-fat diet, ApoE$^{-/-}$ mice were euthanized, and orbital blood was collected. The mouse aortas were examined by staining with Oil Red O, and the aortic root sections were stained with HE and Masson's trichrome. The results indicate that the high-fat diet significantly increased the aortic Oil Red O-positive area (Fig 2A and 2D), atherosclerotic plaque

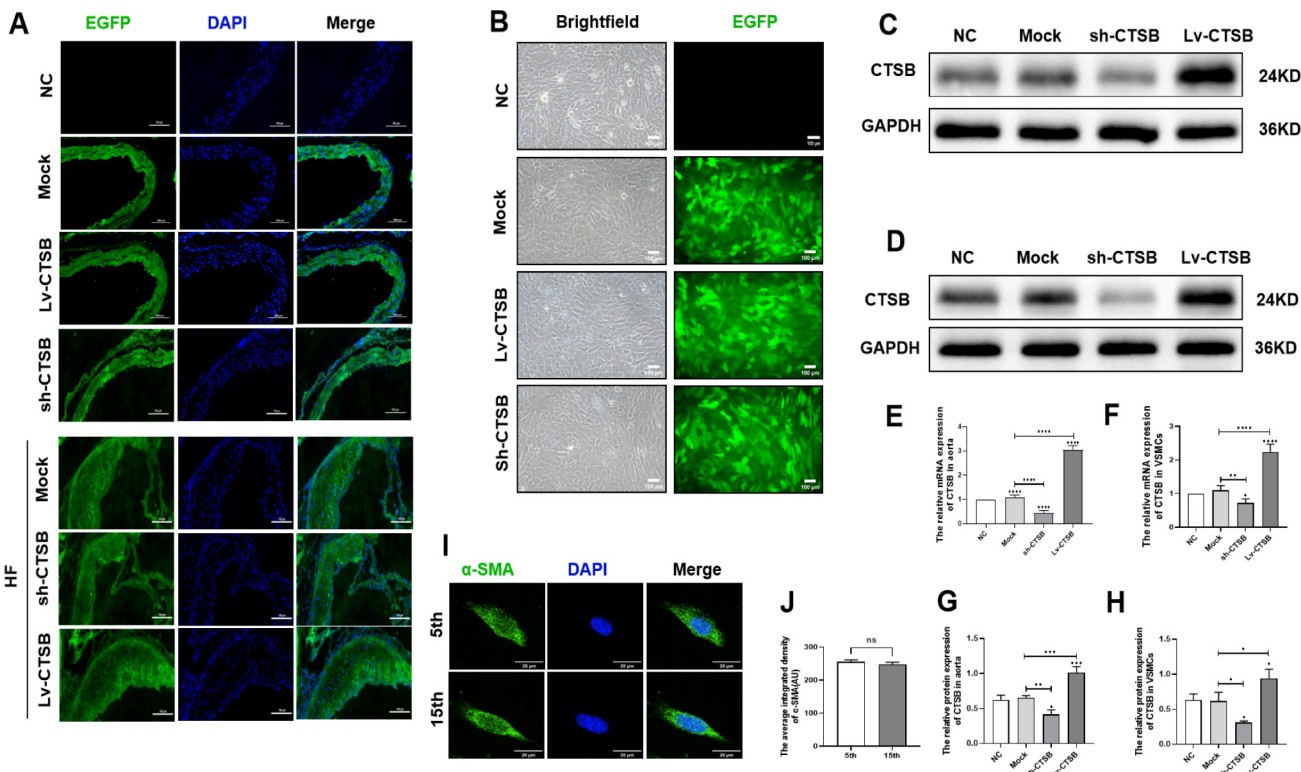

**Fig 1. CTSB overexpression and silencing lentiviruses were successfully transfected into ApoE$^{-/-}$ mouse aortas and VSMCs.** (A) CTSB overexpression and silencing lentiviruses transfected into ApoE$^{-/-}$ mouse aortas (scale bar: 100 μm). (B) CTSB overexpression and silencing lentiviruses transfected into VSMCs (scale bar: 100 μm). (C) Representative western blots of CTSB in aorta. (D) Representative western blots of CTSB in VSMCs. (E) Quantification analysis of CTSB protein level in aorta. (F) Quantification analysis of CTSB protein level in VSMCs.(G) Relative mRNA expression of *CTSB* in aorta. (H) Relative mRNA expression of *CTSB* in VSMCs. (I) The α-SMA expression between 5th and 15th passage of VSMCs. (J) Quantification analysis of α-SMA average integrated density level in VSMCs.Data are presented as means ± SEM. n = 3, *$P<0.05$, **$P<0.01$, ***$P<0.001$, and ****$P<0.0001$. NC: negative control; Mock: empty lentiviral vector; sh-CTSB: CTSB shRNA; Lv-CTSB: CTSB lentivirus; VSMCs: vascular smooth muscle cells; CTSB, cathepsin B.

area (Fig 2B upper and 2E), collagen-positive area (Fig 2B lower and 2F), and serum LDL cholesterol level in the ApoE$^{-/-}$ mice (Fig 2G).

## 3.3 CTSB promotes plaque progression and aggravates pyroptosis of VSMCs in ApoE$^{-/-}$ mice

Oil Red O staining results showed that the Lv-CTSB-transduced mice had a higher positive area in the aorta than the mock-transduced mice (Fig 2A and 2D), with a similar trend in the atherosclerotic plaque area (Fig 2B upper and 2E). Masson's staining showed a lower collagen-positive area in the Lv-CTSB-transduced mice than in the mock-transduced mice (Fig 2B lower and 2F). Conversely, the sh-CTSB-transduced mice displayed the opposite results, indicating that CTSB promoted AS plaque formation.

To further investigate the association between CTSB and pyroptosis pathway factors, we assessed the protein levels of CTSB, NLRP3, caspase-1 p10, and GSDMD-N in the aorta. The levels of these proteins were higher in the Lv-CTSB group and lower in the sh-CTSB group than in the mock group (Fig 2C, 2H–2K).

To investigate the existence of smooth muscle cell pyroptosis in atherosclerotic plaques of ApoE$^{-/-}$ mice and its association with CTSB expression, we conducted immunohistochemical staining for CTSB, NLRP3, caspase-1 p10, GSDMD-N, and the smooth muscle cell marker

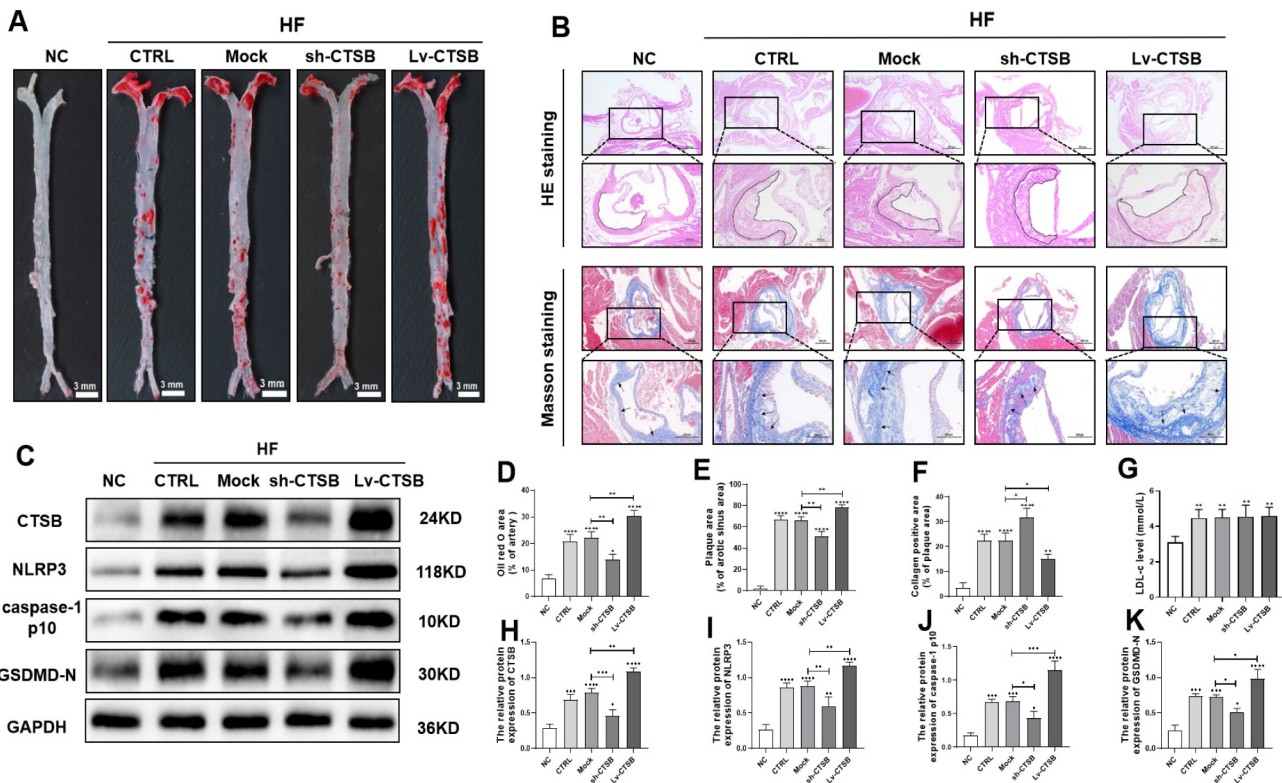

**Fig 2. CTSB overexpression promotes plaque formation and CTSB gene silencing alleviates plaque formation in ApoE$^{-/-}$ mice.** (A) Representative images of Oil Red O staining of aortas (scale bar: 3 mm). (B) HE staining (upper) and Masson's staining (lower) of aortic roots (scale bar: 500 μm). The black box indicates the magnified area shown below each image (scale bar: 200 μm). The black arrows show collagen-positive areas. (C) Representative western blots of CTSB and pyroptosis indicators (NLRP3, cleaved caspase-1 p10 [active caspase-1 subunit], GSDMD-N) in aorta. (D) Quantitative analyses of Oil Red O-positive area in whole aorta. (E) Quantitative analyses of lesion size in aortic roots. (F) Quantitative analyses of collagen area in plaque. (G) LDL-C level. (H–K) Quantification analysis of protein levels in aorta: (H) CTSB, (I) NLRP3, (J) caspase-1 p10, and (K) GSDMD-N. Data are presented as means ± SEM. n = 3, *$P$<0.05, **$P$<0.01, ***$P$<0.001, and ****$P$<0.0001. NC: mice fed chow diet; CTRL: mice fed a high-fat diet; Mock: empty lentiviral vector + high-fat diet; sh-CTSB: CTSB shRNA + high-fat diet; Lv-CTSB: CTSB lentivirus + high-fat diet; CTSB: cathepsin B; HE: hematoxylin and eosin; NLRP3: NOD-like receptor protein 3; GSDMD-N: N-terminal gasdermin D; LDL-C: low-density lipoprotein cholesterol.

protein α-SMA in AS plaques. The Lv-CTSB group exhibited a significant increase in the expression of CTSB, NLRP3, cleaved caspase-1 p10, and GSDMD-N compared with that of the mock group (Fig 3A and 3C–3F). These positive regions were primarily concentrated in the fibrous cap of the plaque and co-localized with the expression of α-SMA. It has been suggested that CTSB aggravates VSMC pyroptosis in AS plaques. There was no significant difference in total α-SMA expression in the aortas of the mice (Fig 3G). In addition, TUNEL and caspase-1 p10 immunofluorescence double staining was used to detect pyroptosis in the plaques (Fig 3B), Revealing that CTSB significantly increased the number of TUNEL- and caspase-1 p10 double-positive cells in the plaques (Fig 3H). These results indicate that pyroptosis, especially VSMC pyroptosis, occurs in atherosclerotic plaques. CTSB promotes plaque progression and aggravates pyroptosis in VSMCs of ApoE$^{-/-}$ mice.

### 3.4 CTSB promotes ox-LDL-induced pyroptosis and NF-κB nuclear translocation in VSMCs

To investigate the role of CTSB in NLRP3-mediated pyroptosis, VSMCs were treated with 150 μg/ml ox-LDL for 24 h. The results show that ox-LDL induced VSMC death (Fig 4A and 4C), LDH release (Fig 4D), and IL-1β secretion (Fig 4E), indicating that ox-LDL promoted

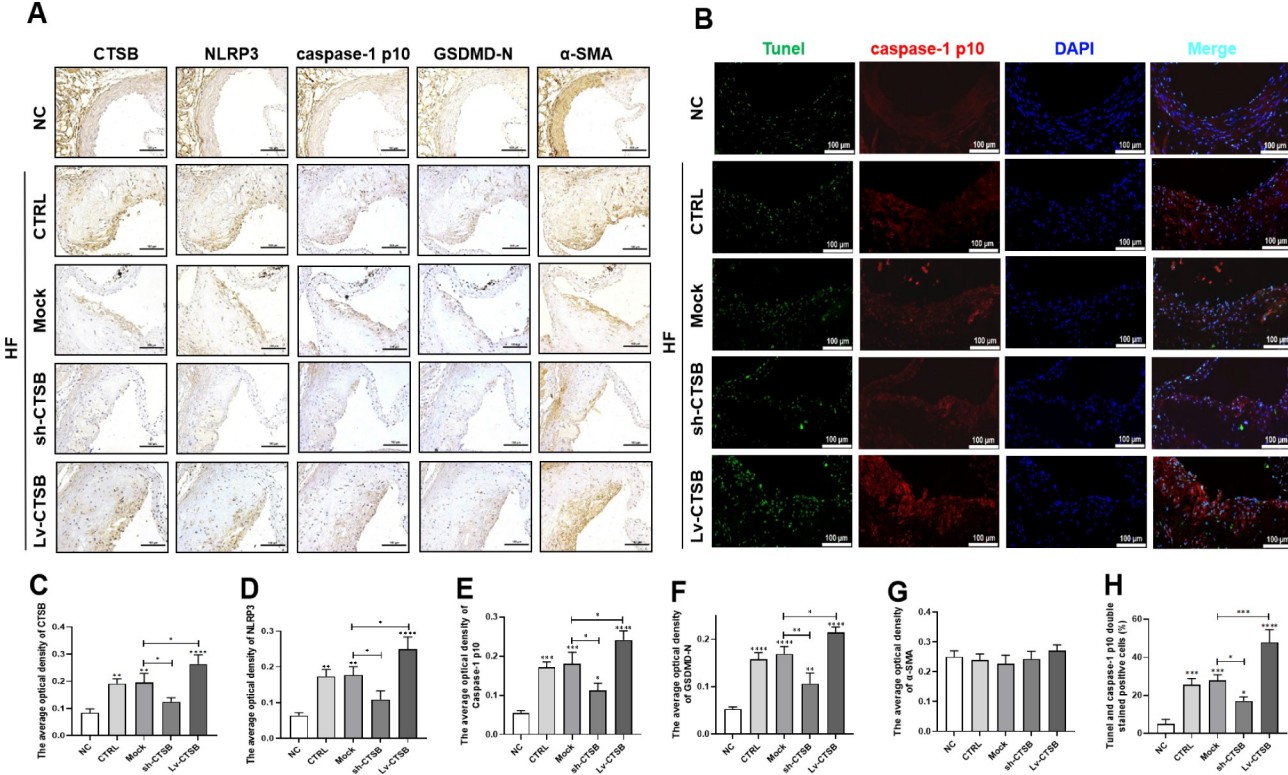

**Fig 3. CTSB overexpression promotes plaque pyroptosis especially VSMC pyroptosis in ApoE$^{-/-}$ mice.** (A) Representative images of immunohistochemistry staining of aortic roots (scale bar: 100 μm). (B) Representative images of TUNEL (green) and caspase-1 p10 (red) immunofluorescence staining of aortic roots (scale bar: 100 μm).(C-G) Quantification analysis of the average optical density in aorta: (C) CTSB, (D) NLRP3, (E) caspase-1 p10, (F) GSDMD-N, (G) α-SMA. (H) Quantification analysis of TUNEL- and caspase-1 p10 double-positive cells. Data are presented as means ± SEM. n = 3, *P<0.05, **P<0.01, ***P<0.001, and ****P<0.0001.n = 3, NC: mice fed chow diet; CTRL: mice fed a high-fat diet; Mock: empty lentiviral vector + high-fat diet; sh-CTSB: CTSB shRNA + high-fat diet; Lv-CTSB: CTSB lentivirus + high-fat diet; HF: high-fat diet; CTSB: cathepsin B; NLRP3: NOD-like receptor protein 3; GSDMD-N: N-terminal gasdermin D; α-SMA: alpha smooth muscle actin; VSMC: vascular smooth muscle cell.

VSMCs pyroptosis. We also found that Lv-CTSB further exacerbated this effect, while sh-CTSB attenuated it. The protein expression levels showed that CTSB promoted ox-LDL-induced VSMCs to express NLRP3, ASC, caspase-1, IL-18, IL-1β, and GSDMD-N, whereas sh-CTSB treatment decreased their levels (Fig 4B and 4A–4G). These results suggest that ox-LDL triggers pyroptosis in VSMCs. Additionally, CTSB overexpression exacerbated and CTSB silencing mitigated ox-LDL-induced pyroptosis.

Our study showed that CTSB promotes the expression of NLRP3; however, the specific mechanism underlying this effect remains unclear. To further investigate, we performed NF-κB-p65 immunofluorescence staining. The results suggest that CTSB promoted NF-κB-p65 nuclear translocation (Fig 5A). In addition, western blotting showed that CTSB promoted the phosphorylation of NF-κB-p65 and IκB-α (Fig 5C, 5E and 5F). These results indicate that CTSB promoted NF-κB/NLRP3 activation and increased pyroptosis in VSMCs.

## 3.5 SN50 inhibits CTSB-mediated NF-κB nuclear translocation and pyroptosis in VSMCs

To investigate the role of CTSB in NF-κB activation, we utilized the NF-κB-p65 nuclear translocation inhibitor SN50, finding that SN50 attenuated the CTSB-increased nuclear

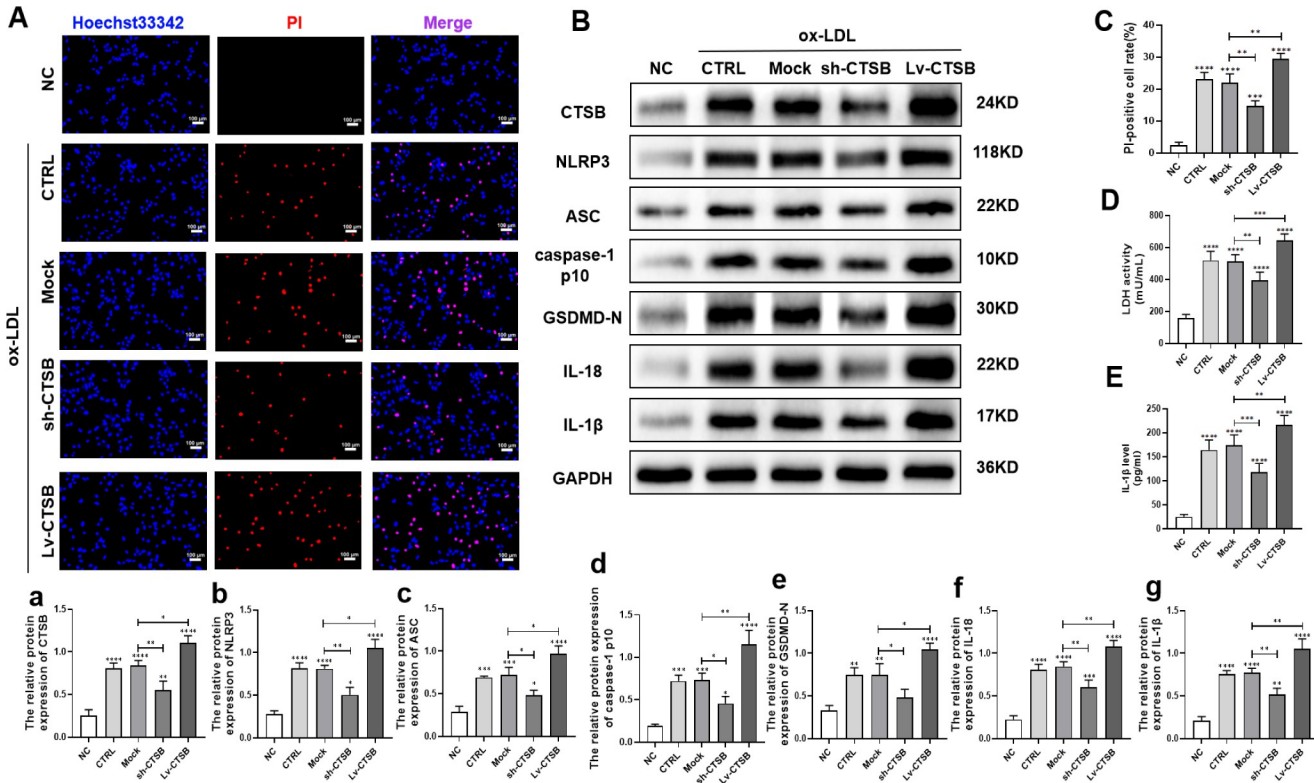

**Fig 4. CTSB overexpression promotes pyroptosis and CTSB gene silencing alleviates pyroptosis in VSMCs.** (A) Representative images of Hoechst 33342 (blue) and propidium iodide (PI, red) staining in VSMCs (scale bar: 100 μm). (B) Representative western blots of CTSB and pyroptosis indicators (NLRP3, ASC, caspase-1 p10 [active caspase-1 subunit], GSDMD-N, IL-18, IL-1β) in VSMCs. (C) Quantification of PI-positive cell percentages. (D) Quantification of lactate dehydrogenase (LDH) activity. (E) Quantification of IL-1β secretion using an enzyme-linked immunosorbent assay. (a–g) Quantification of protein level in VSMCs: (a) CTSB, (b) NLRP3, (c) ASC, (d) caspase-1 p10, (e) GSDMD-N, (f) IL-18, and (g) IL-1β. Data are presented as means ± SEM. n = 3, *$P<0.05$, **$P<0.01$, ***$P<0.001$, and ****$P<0.0001$. NC: VSMCs without any treatment. CTRL: VSMCs treated with 150 μg/ml ox-LDL for 24 h. Mock: empty lentiviral vector + ox-LDL; sh-CTSB: CTSB shRNA + ox-LDL; Lv-CTSB: CTSB lentivirus + ox-LDL; CTSB: cathepsin B; NLRP3: NOD-like receptor protein 3; GSDMD-N: N-terminal gasdermin D; ASC: apoptosis-associated speck-like protein containing a C-terminal caspase recruitment domain; IL: interleukin; ox-LDL: oxidized low-density lipoprotein; VSMCs: vascular smooth muscle cells.

translocation of NF-κB-p65 and levels of p-NF-κB-p65 and p-IκB-α (Fig 5B, 5D and 5G–5H). SN50 also reduced VSMC death (Fig 6A and 6C), LDH release (Fig 6D), and IL-1β secretion (Fig 6E). The protein expression levels of pyroptosis indicators were also reduced by SN50 (Fig 6B, 6F–6K). These results suggest that SN50 reduces NF-κB activation caused by CTSB and alleviates pyroptosis in VSMCs.

## 4. Discussion

In this study, we detected aortic plaques in ApoE[−/−] mice fed a high-fat diet and found that pyroptosis exists in AS plaques, especially in VSMCs. By overexpressing and silencing CTSB in the ApoE[−/−] mouse AS model, we confirmed that CTSB promotes VSMCs pyroptosis in atherosclerotic plaques and contributes to the progression of AS. These findings support the hypothesis that CTSB participates in the progression of AS by promoting pyroptosis of VSMCs. In our *in vitro* experiments, we observed that ox-LDL treatment significantly enhanced VSMC pyroptosis. We found that CTSB plays a dual role in promoting VSMC pyroptosis. On one hand, it activated the NLRP3 inflammasome, and on the other hand, it stimulated NF-κB activation, leading to the upregulation of NLRP3 expression. However, we discovered that SN50 treatment effectively mitigated the impact of CTSB on VSMC pyroptosis, clarifying the

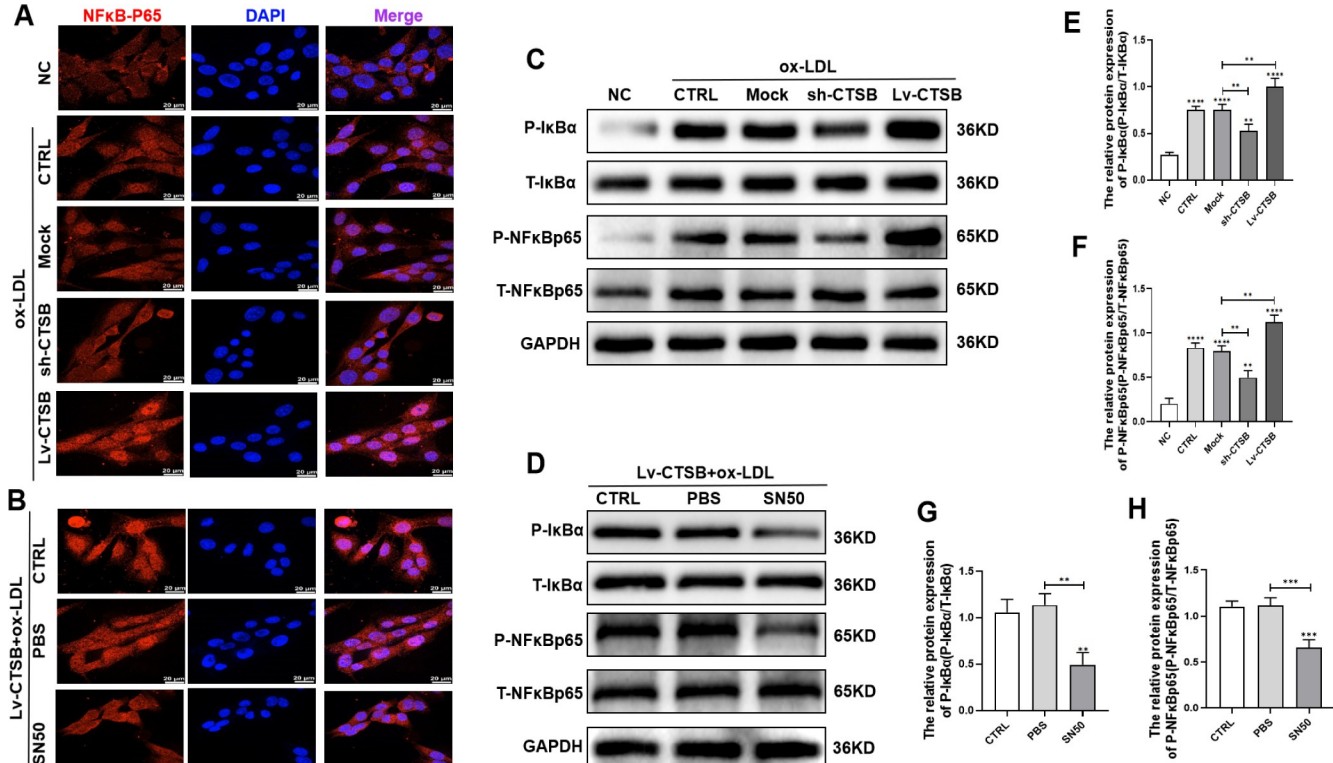

**Fig 5. CTSB promotes pyroptosis through NF-κB signaling in VSMCs.** (A) Representative images of NF-κB-p65 (red) immunofluorescence and DAPI (blue) staining of ox-LDL-treated VSMCs (scale bar: 20 μm). NC: VSMCs without any treatment; CTRL: VSMCs treated with 150 μg/ml ox-LDL for 24 h. Mock: empty lentiviral vector + ox-LDL; sh-CTSB: CTSB shRNA + ox-LDL; Lv-CTSB: CTSB lentivirus + oxLDL. (B) Representative images of NF-κB-p65 (red) immunofluorescence and DAPI (blue) staining of Lv-CTSB- and ox-LDL-treated VSMCs (scale bar: 20 μm). CTRL: Lv-CTSB + ox-LDL; PBS: Lv-CTSB + ox-LDL + PBS; SN50: Lv-CTSB + ox-LDL + SN50 (75 μg/ml). (C) Representative western blots of IκB-α and NF-κB-p65 in ox-LDL-treated VSMCs. (D) Representative western blots of IκB-α and NF-κB-p65 in Lv-CTSB + ox-LDL–treated VSMCs. (E–H) Quantification of protein levels in VSMCs: (E) p-IκBα in ox-LDL-treated VSMCs, (F) p-NF-κB-p65 in ox-LDL-treated VSMCs, (G) p-IκBα in Lv-CTSB + ox-LDL-treated VSMCs, and (H) p-NF-κB-p65 in Lv-CTSB + ox-LDL-treated VSMCs. p-IκB-α: phosphorylated IκB-α; T-IκB-α: total IκB-α); p-NF-κB-p65: phosphorylated NF-κB-p65; T-NF-κB-p65: total NF-κB-p65. Data are presented as means ± SEM. n = 3, *$P<0.05$, **$P<0.01$, ***$P<0.001$, and ****$P<0.0001$.ox-LDL: oxidized low-density lipoprotein; CTSB: cathepsin B; VSMCs: vascular smooth muscle cells.

mechanism by which CTSB-mediated VSMC pyroptosis promotes AS, and providing a theoretical basis for the participation of VSMC pyroptosis in AS pathology.

AS is a chronic inflammatory disease characterized by lipid deposition in the arteries. Cell death and inflammation occur during the development of AS [23]. VSMCs and their secreted ECM are not only an important part of AS plaques, but also an important criterion for judging the stability of AS plaques [24]. Pan et al. [25] and Li et al. [26] confirmed the existence of pyroptosis in AS. Specifically, they found that NLRP3-mediated VSMC pyroptosis promoted the formation and progression of AS plaques in ApoE[-/-] mice. In the present study, ApoE[-/-] mice were fed a high-fat diet to induce AS. The results showed that the high-fat diet promoted the formation of AS plaques in these mice. Additionally, we utilized TUNEL and caspase-1 p10 immunofluorescence staining to demonstrate the presence of pyroptosis in AS plaques, and immunohistochemical staining revealed that the distribution of CTSB, NLRP3, caspase-1 p10, and GSDMD-N was consistent with that of the VSMC marker α-SMA, particularly on the plaque surface and fibrous cap. This suggests that VSMC pyroptosis is involved in the pathogenesis of AS.

Previous research has indicated that CTSB plays a role in the development of AS by inducing endothelial and vascular smooth muscle cell apoptosis and promoting foam cell formation

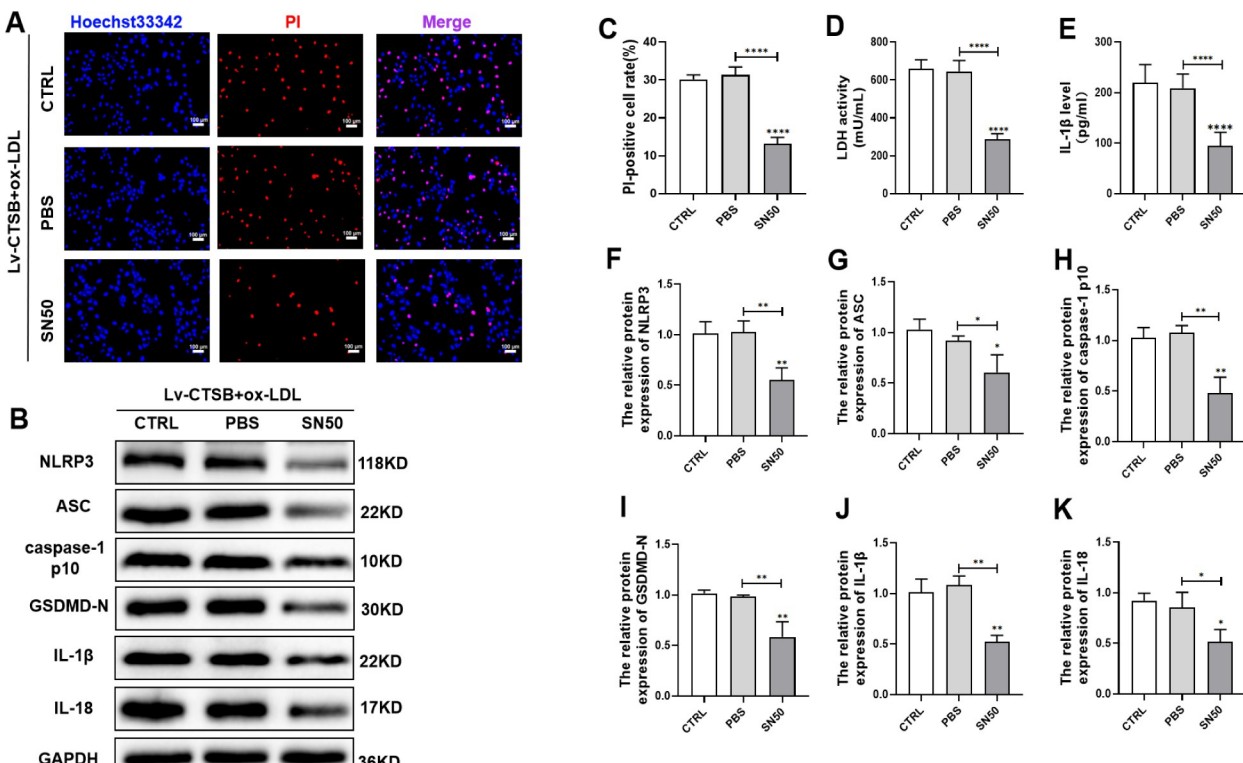

**Fig 6. SN50 alleviates CTSB-mediated pyroptosis in VSMCs.** (A) Representative images of Hoechst 33342 (blue) and propidium iodide (PI, red) staining in VSMCs (scale bar: 100 μm). (B) Representative western blots of pyroptosis indicators (NLRP3, ASC, cleaved caspase-1 p10 [active caspase-1 subunit], GSDMD-N, IL-18, IL-1β) in VSMCs. (C) Quantification of PI-positive cell percentages. (D) Quantification of lactate dehydrogenase (LDH) activity. (E) Quantification of IL-1β secretion using an enzyme-linked immunosorbent assay. (F–H) Quantification of protein level in VSMCs: (F) NLRP3, (G) ASC, (H) caspase-1 p10, (I) GSDMD-N, (J) IL-1β, and (K) IL-18. Data are presented as means ± SEM. n = 3, *P<0.05, **P<0.01, ***P<0.001, and ****P<0.0001. CTRL: Lv-CTSB + ox-LDL; PBS: Lv-CTSB + ox-LDL + PBS; SN50: Lv-CTSB + ox-LDL + SN50 (75 μg/ml); CTSB: cathepsin B; NLRP3: NOD-like receptor protein 3; GSDMD-N: N-terminal gasdermin D; ASC: apoptosis-associated speck-like protein containing a C-terminal caspase recruitment domain; IL: interleukin; ox-LDL: oxidized low-density lipoprotein; VSMCs: vascular smooth muscle cells.

[27–29]. Jia et al. [30] demonstrated that CTSB promotes pyroptosis in human umbilical vein endothelial cells by activating the NLRP3 inflammasome. This activation was found to exacerbate inflammatory infiltration of the coronary artery in mice. Chevriaux et al. [31] used CTSB−/− bone marrow-derived macrophages to demonstrate that CTSB interacts with NLRP3 at the endoplasmic reticulum level, leading to caspase-1 activation. Progressively more evidence has shown that CTSB mediates pyroptosis by activating NLRP3. However, it is still uncertain whether CTSB is involved in the activation of NLRP3-mediated VSMC pyroptosis in AS. In this study, ApoE−/− mice were transduced with lentiviruses to overexpress or knockdown CTSB in the aorta. We found that overexpression of CTSB resulted in increased levels of serum IL-1β and LDH in ApoE−/− mice, leading to activation of caspase-1 and expression of GSDMD-N in the aorta. Furthermore, CTSB overexpression promoted VSMC pyroptosis in AS plaques and aggravated plaque progression. Silencing CTSB had the opposite effect. The levels of LDL cholesterol in ApoE−/− mice were not affected by either the overexpression or silencing of CTSB. This suggests that the mechanism by which CTSB promotes AS plaques may not be related to changes in blood lipid levels but rather to the promotion of inflammation. The above results indicate that CTSB is involved in the development of AS by activating NLRP3-mediated VSMC pyroptosis. Furthermore, we found that CTSB not only enhanced the activation of NLRP3 in ApoE−/− mice, but also increased the expression of NLRP3.

The NF-κB pathway is a classical inflammatory signaling pathway. NF-κB is present in the cytoplasm as a p50 or p65 homodimer or as a p50/p65 heterodimer; the heterodimer is the most prevalent [22]. Translocation of the NF-κB dimer from the cytoplasm to the nucleus can promote the transcription of genes and expression of proteins such as NLRP3, IL-18, and IL-1β [32]. Activation of the NF-κB signaling pathway provides the necessary components for NLRP3 inflammasome-mediated pyroptosis. SN50 is a synthetic peptide that competitively binds to the NF-κB p50 subunit. This binding inhibits the nuclear translocation of NF-κB and blocks the activation of the NF-κB dimer [33]. In comparison to JSH-23, a polypeptide that binds to the NF-κB-p65 subunit, SN50 does not affect the transcription and synthesis of IL-1β. Therefore, SN50 is more suitable for studying the effects of NF-κB on pyroptosis [34]. Studies conducted by Pan et al. [25] and Li et al. [26] confirmed the involvement of ox-LDL in VSMC pyroptosis. To induce VSMC pyroptosis *in vitro*, a concentration of 150 μg/ml ox-LDL was utilized. We found that CTSB overexpression led to an increase in the number of PI-positive cells and the levels of IL-1β and LDH in the cell culture medium. Additionally, the levels of NLRP3, caspase-1 p10, IL-18, IL-1β, and GSDMD-N increased with CTSB overexpression, whereas CTSB silencing had the opposite effect. These findings suggest that CTSB plays a role in promoting ox-LDL-induced VSMC pyroptosis by activating the NLRP3 inflammasome. Liu et al. [35] discovered that CTSB contributes to doxorubicin-induced myocardial injury via the NF-κB signaling pathway. Their research revealed that CTSB promotes the phosphorylation of NF-κB-p65 and its nuclear translocation, leading to increased cardiomyocyte apoptosis. Interestingly, in our *in vitro* experiments, we also observed that ox-LDL promoted the phosphorylation and nuclear translocation of NF-κB-p65. CTSB overexpression worsened VSMC pyroptosis, whereas CTSB silencing improved it. These findings indicate that CTSB may exacerbate VSMC pyroptosis by enhancing NF-κB activation and nuclear translocation, leading to the upregulation of NLRP3 expression. We used ox-LDL to induce VSMC pyroptosis via overexpression of CTSB and then administered the NF-κB inhibitor SN50, which effectively reduced the NF-κB activation and nuclear translocation caused by CTSB overexpression. As a result, we observed a decrease in NLRP3 protein expression and a reduction in VSMC pyroptosis. The above results indicate that CTSB not only mediates VSMC pyroptosis by activating NLRP3, but also aggravates VSMC pyroptosis by promoting NLRP3 expression.

In summary, our results elucidated the important role of CTSB in NLRP3 inflammasome-mediated VSMC pyroptosis and AS. Targeted inhibition of CTSB-activated NF-κB to reduce VSMC pyroptosis and vascular inflammation may be a promising treatment for atherosclerotic diseases.

## Supporting information

**S1 Raw images.**
(PDF)

## Author Contributions

**Data curation:** Quanwei Zhao, Fujun Liao, Long Chen.

**Formal analysis:** Quanwei Zhao, Bo Zhou, Fujun Liao.

**Funding acquisition:** Danan Liu.

**Investigation:** Danan Liu.

**Methodology:** Hui Li.

**Project administration:** Danan Liu.

**Software:** Hui Li, Bo Zhou.

**Validation:** Hui Li.

**Visualization:** Quanwei Zhao.

**Writing – original draft:** Hui Li.

**Writing – review & editing:** Danan Liu.

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
