## [Decision Letter · Decision Letter 0]

17 Jul 2023

PONE-D-23-17730Cathepsin B aggravates atherosclerosis in ApoE deficient mice by modulating

VSMCs pyroptosis  through NF-κB / NLRP3 signaling pathwayPLOS ONE

Dear Dr. Liu,

Thank you for submitting your manuscript to PLOS ONE. After careful consideration, we feel that it has merit but does not fully meet PLOS ONE’s publication criteria as it currently stands. Therefore, we invite you to submit a revised version of the manuscript that addresses the points raised during the review process.

We look forward to receiving your revised manuscript.

Kind regards,

Lingshan Gou, Ph.D.

Academic Editor

PLOS ONE

Journal Requirements:

   "Danan Liu plays a role in study design,decision to publish and preparation of the manuscript.Danan Liu provide funding support.This study was supported by the Guizhou Science and Technology Innovation Talent Team Project [Guizhou Science and Technology Cooperation Platform Talent (2020)5014]; Guizhou Province "Hundred" level innovative talents Training Program project [Guizhou Science and Technology Cooperation Talents (2015) No. 4026]."

7. PLOS ONE now requires that authors provide the original uncropped and unadjusted images underlying all blot or gel results reported in a submission’s figures or Supporting Information files. This policy and the journal’s other requirements for blot/gel reporting and figure preparation are described in detail at https://journals.plos.org/plosone/s/figures#loc-blot-and-gel-reporting-requirements and https://journals.plos.org/plosone/s/figures#loc-preparing-figures-from-image-files. When you submit your revised manuscript, please ensure that your figures adhere fully to these guidelines and provide the original underlying images for all blot or gel data reported in your submission. See the following link for instructions on providing the original image data: https://journals.plos.org/plosone/s/figures#loc-original-images-for-blots-and-gels. 

8. Please ensure that you refer to Figure 4 in your text as, if accepted, production will need this reference to link the reader to the figure.

Reviewers' comments:

Reviewer's Responses to Questions

**Comments to the Author**

1. Is the manuscript technically sound, and do the data support the conclusions?

Reviewer #1: Yes

Reviewer #2: Partly

2. Has the statistical analysis been performed appropriately and rigorously? 

Reviewer #1: Yes

Reviewer #2: No

3. Have the authors made all data underlying the findings in their manuscript fully available?

Reviewer #1: Yes

Reviewer #2: No

4. Is the manuscript presented in an intelligible fashion and written in standard English?

Reviewer #1: Yes

Reviewer #2: No

5. Review Comments to the Author

Reviewer #1: This manuscript by Li et al., describes their investigation of the role of cathepsin B (CTSB) in pyroptosis (i.e., inflammatory-mediated cell death) and atherosclerosis.

They utilize ApoE-/- high fat diet mouse model of atherosclerosis, and mouse aortic vascular smooth muscle cells in vitro, and perform gain and loss of CTSB function experiments (using lentiviral vectors to either overexpress or knockdown expression of CTSB in vivo and in vitro; Fig 1).

Their data show that

CTSB overexpression causes pyroptosis (as indicated by increased NLRP3, Caspase-1 p10, and GSDMD-N) and increases plaque formation in ApoE-/- hfd mice, whereas CTSB silencing reduces these (Fig 2 and 3).

CTSB overexpression promotes oxLDL-induced pyrotosis in VSMC, and CTSB silencing reduces it (Fig 4).

SN50 (an NF-kB-p65 nuclear translocation inhibitor) treatment inhibited CTSB-induced pyroptosis in VSMC. (Fig 5)

CTSB promotes pyroptosis via Nf-kB signaling (Fig 6)

They conclude that CTSB promotes VSMC pyroptosis, and therefore atherogenesis, by activating the NLRP3 inflammasome via Nf-kB activation.

In general, the data are convincing and supportive of their conclusions.

Main points:

1. Several corrections are needed in the text including spelling errors (e.g., in Methods section: H&E and Masson STAINING, Immunohistochemistry STAINING), and lack of proper spacing between sentences in multiple places throughout.

2. In figure legends the following corrections are necessary-

- Fig 1 legend: E F G H are incorrectly referenced with respect to the order of the protein and mRNA data presented.

- Fig 2 legend: ‘G’ is LDL data (this is omitted); should be h-k, not g-j

3. Fig 4 legend : remove the first sentence as not necessary.

4. Results section goes from Fig 4 to Fig 6 , with no mention of Fig 5. Need to include.

5. Discussion: Should start with a clear statement of the findings of this study.

6. Final sentence in the Discussion should read “Targeted inhibition of CTSB-ACTIVATED Nf-kB…..

Reviewer #2: In this report Li et al used the ApoE knockout mouse model to investigate the role of cathepsin in aortic atherosclerosis formation and cultured vascular smooth muscle cells death. The authors specifically measured the effects of cathepsin overexpression or know down (using shRNA) on specific markers of inflammatory cell death or pyroptosis in whole aorta or cultured VSMCs. The markers were measured by IHC, Western blot, or RT-PCR. The authors concluded that cathepsin contributes to the development of arthersclerotic plaque and pyroptosis in ApoE knockout mice subjected to high fat diet.

In the study, APE knockout mice were put on regular or high fat diet at the age of 8-weeks and kept on those diets till they reach the age of 20 weeks. Some groups of mice were injected with lentivirus overexpression cathepsin or sh-RNA to silence the expression of cathepsin for the duration of the study. The efficiency of transfections was measured two weeks after the construct delivery. At 20 weeks of age, all mice were sacrificed, and aortic tissue were subjected to different assays to measure the markers of pyroptosis and aortic AS progression.

The authors also studies the role of NFkB pathway in mediated cathepsin-induced pyroptosis using cultured mouse aortic VSMCs.

The study is well-designed and the rationale for methodological approach is clear in most cases. However, I have a few major and minor comments and questions that I think should be addressed before the paper is acceptable for publications:

Major Comments:

1) In the introduction of the paper, it is not clear why authors decided to study cathepsin. There are many other signaling molecules that are thought to be related to SMCs death, inflammatory responses, and atherosclerosis development. The authors need to add more rationale in the introduction to guide their readers. The authors need to first establish the potential link between cathepsin and AS before delving further into the mechanism of cathepsin contribution within the context of AS.

2) Have the authors compared the contribution of apoptosis vs. pyroptosis in their study model?

3) What was the reason to only use male mice? Why didn’t the authors include female mice in their study? If the authors are interested in clinical relevance of their findings, the inclusion of female mice and the impact of female sex hormone in the inflammatory process of pyroptosis seems to be important.

4) The authors do not make the definition of the “Mock” group clear. Is the mock group infected with the GFP tagged neutral construct?

5) The authors check the transfection efficiency 2 weeks after the construct delivery. Did the author also measure the construct expression in the mouse aorta at the end of 20 weeks and towards the end of experiments? Do they expect to see any drop in the expression level of those constructs?

6) When measuring the efficiency of transfection, how the authors make sure that the detected green fluorescence is not coming from aortic wall elastin fibers. Elastin fibers are known to emit autofluorescence. How did the author make sure that the signal is specific to their constructs? Besides, the author did not include any image of non-transfected (NC) mice aorta as the negative control in figure 1A, so it is very hard to make sure that the green signal is due to transfection and not from non-specific elastin auto-fluorescence.

7) Figure 1A: when comparing the EGFP column for sh-CTSB vs. Lv-CTSB, there seems to be obvious differences in expression pattern. What could be the reason for this observation?

8) I have major concerns regarding all provided histological images. The authors are not clear on how the tissue sections were prepared (e.g., serial sections from the exact aortic area or random cross sections collected from different regions). They do not provide any clear description on how exactly these serial or non-serial sections were analyzed and if they had ROIs for each section. Also, there are no negative controls included in those image panels, so it is hard to determine how specific their staining is. Also, there is no quantification and actual statistical values and analysis provided for those immunohistochemistry or conventional histological staining provided throughout the figures.

9) The authors used cultured VSMCs within passages 5-15 for their experiments. Considering the high number of passages of SMCs after the initial seeding, did the author check those cells for alpha-actin SMC to ensure that the cultured cells still show the SMC phenotype?

10) Figure 2: It is recommended to provide a magnified image of positively stained areas and use arrows to mark them.

11) Figure 3A & 3B: aortic tissue cross sections presented in figure 3A and 3B do not correlate. Are these sections prepared from the same aortic sections?

12) In figure 3, the authors only provide images of IHC for markers of pyroptosis. However, they have not performed any image and statistical analysis to quantify the expression levels for those markers. Despite the lack of analysis, at the end of sub-section 3.4, the authors concluded that “CTSB aggravated pyroptosis of VSMCs in APOE-/- mice plaques”. I don’t believe that the provided evidence warrants such conclusions.

13) I couldn’t find any reference in the results section to data presented in Figure 5.

14) The authors claimed that SN50 treatment leads to decrease phosphorylation of IKB. To my knowledge, SN50 blocks the translocation and therefore activation of NFKBp65 into the nucleus. What is the relationship between SN50 mechanism of action and the pathway controlling IKB phosphorylation in the cytoplasm?

Minor Comments:

1) The manuscript needs some re-writing in proper scientific English language. Throughout the manuscripts there are so many errors with respect the proper English words or phrase and spacing in the beginning of sentences or before the brackets.

2) When referring to other authors throughout the manuscript use the proper reference (e.g., Li et. al)

3) In the text, all “in vitro” or “in vivo” phrases should be in Italics.

4) Provide the full name of all markers and compounds for the first time in your text. Some of the abbreviated words are not known to all readers.

5) Throughout the manuscript there are many phrases that need to be corrected and revised. For example, “antigen repair” should be replaced with “antigen retrieval”, “cell-attached slides” with “tissue cross sections”, etc….

6) There seem to be redundancies in subsection 3.3 & 3.4 under the results section. I suggest the authors combine these two subsections.

7) Figure 3: on the first line of legend, “plaue pyropsis” should be corrected to “plaque pyroptosis”.

6. PLOS authors have the option to publish the peer review history of their article (what does this mean?). If published, this will include your full peer review and any attached files.

Reviewer #1: No

Reviewer #2: No

---

## [Author Response · Author response to Decision Letter 0]

29 Aug 2023

Response to academic editor :

Journal Requirements:

Response:We have revised the manuscript according to the requirements of PLOS ONE style templates.

Response:We have made a comprehensive examination of the language problems of manuscripts,and the manuscript had be reviewed and edited by language services of Editage.

Response:We have deleted the funding information from the manuscript.

4.Please state what role the funders took in the study. Please include this amended Role of Funder statement in your cover letter; we will change the online submission form on your behalf.

Response:We have revised the following information in cover letter :"The funders had no role in study design, data collection and analysis, decision to publish, or preparation of the manuscript." 

5.In your Data Availability statement, you have not specified where the minimal data set underlying the results described in your manuscript can be found. "Upon re-submitting your revised manuscript, please upload your study’s minimal underlying data set as either Supporting Information files or to a stable, public repository and include the relevant URLs, DOIs, or accession numbers within your revised cover letter. 

Response:Our study’s minimal underlying data can be found at Supporting Information files。

6.We note that you have stated that you will provide repository information for your data at acceptance. Should your manuscript be accepted for publication, we will hold it until you provide the relevant accession numbers or DOIs necessary to access your data. If you wish to make changes to your Data Availability statement, please describe these changes in your cover letter and we will update your Data Availability statement to reflect the information you provide.

Response:We have described the Data Availability statement in cover letter.

7.PLOS ONE now requires that authors provide the original uncropped and unadjusted images underlying all blot or gel results reported in a submission’s figures or Supporting Information files. 

Response:Our study 's blot results can be found in Supporting Information files.

8. Please ensure that you refer to Figure 4 in your text as, if accepted, production will need this reference to link the reader to the figure.

Response:It is our negligence of no mention of Fig4 in results section,we are deeply sorry about this.In order to ensure the logic of the text, we rewrite the results 3.3 to 3.5, we adjust the order of figure, and check the reference to the figure lend in the results section.

*Response to Reviewers:

Reviewer #1:

Main points:

1. Several corrections are needed in the text including spelling errors (e.g., in Methods section: H&E and Masson STAINING, Immunohistochemistry STAINING), and lack of proper spacing between sentences in multiple places throughout.

Response:We are very sorry for our incorrect writing. We have made correction according to the Reviewer 's comments and adjusted the text spacing appropriately.

2. In figure legends the following corrections are necessary-

- Fig 1 legend: E F G H are incorrectly referenced with respect to the order of the protein and mRNA data presented.

- Fig 2 legend: ‘G’ is LDL data (this is omitted); should be h-k, not g-j

Response:We are very sorry for our negligence of figure legends reference.We have revised the notes in Figure 1 and Figure 2 according to the reviewer 's recommendations.

3. Fig 4 legend : remove the first sentence as not necessary.

Response:Indeed, like the reviewer 's suggestion, the first sentence of Fig4 is redundant, and we have deleted the first sentence.

4. Results section goes from Fig 4 to Fig 6 , with no mention of Fig 5. Need to include.

Response:We are very sorry for our negligence of no mention of Fig5 in results section.In order to ensure the logic of the text, we rewrite the results 3.3 to 3.5, we adjust the order of figure, and check the reference to the figure lend in the results section.

5. Discussion: Should start with a clear statement of the findings of this study.

Response:Thanks for the reviewer’s kind reminds,We have rewritten the first paragraph of the discussion to state the findings of this study.

6. Final sentence in the Discussion should read “Targeted inhibition of CTSB-ACTIVATED Nf-kB…..

Response:We are very sorry for our incorrect writing. We have made correction according to the Reviewer 's comments.

Reviewer #2: 

Major Comments:

1) In the introduction of the paper, it is not clear why authors decided to study cathepsin. There are many other signaling molecules that are thought to be related to SMCs death, inflammatory responses, and atherosclerosis development. The authors need to add more rationale in the introduction to guide their readers. The authors need to first establish the potential link between cathepsin and AS before delving further into the mechanism of cathepsin contribution within the context of AS.

Response:It is really true as Reviewer suggested that we didn’t clarified why we decided to study cathepsin.We have added the literatue review of the relationship between cathepsin B and atherosclerosis in the introduction, and established the potential link between cathepsin B and atherosclerosis.

2) Have the authors compared the contribution of apoptosis vs. pyroptosis in their study model?

Response:Apoptosis and pyroptosis were not compared in the model in this study. Although both belong to programmed cell death, their mechanisms and detection methods are not the same. According to the the mechanism of pyroptosis, the occurrence of pyroptosis was determined by measuring the expression of pyroptosis pathway proteins, the expression of LDH, IL-18, IL-1β, and Tunel / caspae-1 immunofluorescence co-staining.

3) What was the reason to only use male mice? Why didn’t the authors include female mice in their study? If the authors are interested in clinical relevance of their findings, the inclusion of female mice and the impact of female sex hormone in the inflammatory process of pyroptosis seems to be important.

Response:As far as we know, sex hormones play an important role in the development of atherosclerosis. As the reviewer reminds, estrogen is important in the process of inflammation promoting atherosclerosis.Some studies have shown that oestrogen decreases adhesion molecule expression, decreases neutrophil infiltration and pro-inflammatory cytokines such as TNFα, IL-1β and IL-6 secretion and therefore halting atherosclerosis in female mice.[1,2].Previous studies have shown that oestrogen is atheroprotective in young women and in ovariectomized female mice and rats treated with oestrogen due to anti-inflammatory and vasoprotective effects.[3,4,5] However, in the most frequently used atherosclerotic mouse models, Apolipoprotein E knockout (Apoe−/−) and low-density lipoprotein receptor knockout (Ldlr−/−), female mice up to 6months old, fed normal chow or atherosclerotic diet, have in general a greater atherosclerotic lesion size and overall burden compared with their male counterparts.In 8+-month-old ApoE−/− mice on normal chowor atherosclerotic diet, males were reported to have equal or larger plaques.[6,7]The animal model used in this study was Apolipoprotein E knockout (Apoe−/−) mice, which were studied until they reached 20 weeks of age,which is less than 6 months.Considering that gender differences could potentially impact the experimental results, we decided to only utilize male mice.

Reference for the response are as following:

1.Arenas IA, Armstrong SJ, Xu Y, Davidge ST. Chronic tumor necrosis factor-alpha inhibition enhances NO modulation of vascular function in estrogen-deficient rats. Hypertension. 2005 Jul;46(1):76-81. doi: 10.1161/01.HYP.0000168925.98963.ef. Epub 2005 May 23. PMID: 15911738.

2.Evans BR, Yerly A, van der Vorst EPC, Baumgartner I, Bernhard SM, Schindewolf M, Döring Y. Inflammatory Mediators in Atherosclerotic Vascular Remodeling. Front Cardiovasc Med. 2022 May 4;9:868934. doi: 10.3389/fcvm.2022.868934. PMID: 35600479; PMCID: PMC9114307.

3.Bakir S, Mori T, Durand J, Chen YF, Thompson JA, Oparil S. Estrogen-induced vasoprotection is estrogen receptor dependent: evidence from the balloon-injured rat carotid artery model. Circulation. 2000 May 23;101(20):2342-4. doi: 10.1161/01.cir.101.20.2342. PMID: 10821807.

4.Miller AP, Feng W, Xing D, Weathington NM, Blalock JE, Chen YF, Oparil S. Estrogen modulates inflammatory mediator expression and neutrophil chemotaxis in injured arteries. Circulation. 2004 Sep 21;110(12):1664-9. doi: 10.1161/01.CIR.0000142050.19488.C7. Epub 2004 Sep 7. PMID: 15353495.

5.Miller AP, Xing D, Feng W, Fintel M, Chen YF, Oparil S. Aged rats lose vasoprotective and anti-inflammatory actions of estrogen in injured arteries. Menopause. 2007 Mar-Apr;14(2):251-60. doi: 10.1097/01.gme.0000235366.39726.f6. PMID: 17194962.

6.Yerly A, van der Vorst EPC, Baumgartner I, Bernhard SM, Schindewolf M, Döring Y. Sex-specific and hormone-related differences in vascular remodelling in atherosclerosis. Eur J Clin Invest. 2023 Jan;53(1):e13885. doi: 10.1111/eci.13885. Epub 2022 Oct 22. PMID: 36219492.

7.Man JJ, Beckman JA, Jaffe IZ. Sex as a Biological Variable in Atherosclerosis. Circ Res. 2020 Apr 24;126(9):1297-1319. doi: 10.1161/CIRCRESAHA.120.315930. Epub 2020 Apr 23. PMID: 32324497; PMCID: PMC7185045.

4) The authors do not make the definition of the “Mock” group clear. Is the mock group infected with the GFP tagged neutral construct?

Response:We are very sorry for our negligence of the definition of the “Mock” group.The mock group is defined as infected with the GFP tagged neutral construct.According to the reviwer’s reminds , We have defined the Mock group in Materials and Methods 2.2 and 2.10.

5) The authors check the transfection efficiency 2 weeks after the construct delivery. Did the author also measure the construct expression in the mouse aorta at the end of 20 weeks and towards the end of experiments? Do they expect to see any drop in the expression level of those constructs?

Response:Thanks for the reviewer’s kind reminds,We also detected the expression of mouse aortic construct at the 20 th week. Due to the limitation of space, we only provided the expression picture of CTSB protein in Fig2 c, and the results showed that the expression of CTSB still existed at the 20 th week. We also detected EGFP in the aorta of mice. The results showed that plaques appeared in the aorta of mice fed with virus transfection combined with high-fat diet, and EGFP green fluorescence existed in the plaques.The picture is as follows :

6) When measuring the efficiency of transfection, how the authors make sure that the detected green fluorescence is not coming from aortic wall elastin fibers. Elastin fibers are known to emit autofluorescence. How did the author make sure that the signal is specific to their constructs? Besides, the author did not include any image of non-transfected (NC) mice aorta as the negative control in figure 1A, so it is very hard to make sure that the green signal is due to transfection and not from non-specific elastin auto-fluorescence.

Response:Thanks for the reviewer’s professionnal reminds,It is really true as reviewer mentioned that elastin fibers could emit autofluorescence.So,we not only have added non-transfected (NC) mice aorta image in figure1A but also non-transfected (NC) VSMCs image in figure 1B to increase the reliability of the results.

7) Figure 1A: when comparing the EGFP column for sh-CTSB vs. Lv-CTSB, there seems to be obvious differences in expression pattern. What could be the reason for this observation?

Response:We re-examined the original data of the Fig1A sh-CTSB group and found that as observed by the reviewer, the expression of EGFP in this group seemed to be less expressed in the cytoplasm and more expressed in the elastic fibers. So we re-stained the sections of the same part of this group, and the results showed that the expression of green fluorescence increased. We speculate that the cause of this situation is related to fluorescence quenching. We re-replaced the image of the sh-CTSB group in Fig1A.

8) I have major concerns regarding all provided histological images. The authors are not clear on how the tissue sections were prepared (e.g., serial sections from the exact aortic area or random cross sections collected from different regions). They do not provide any clear description on how exactly these serial or non-serial sections were analyzed and if they had ROIs for each section. Also, there are no negative controls included in those image panels, so it is hard to determine how specific their staining is. Also, there is no quantification and actual statistical values and analysis provided for those immunohistochemistry or conventional histological staining provided throughout the figures.

Response:We are deeply sorry for our unprofessional handling of histological images.The sections are serial sections from the aortic root area.All histological images were based on the aortic root aortic valve as a reference, and the plaque at the beginning of the valve was used as ROIs.We added magnified images to each group of HE and Masson stained images. Plaque positive areas and ROIs were depicted with black lines on the image.The positive area of masson staining was marked with black arrows.We are very sorry for our negligence of data provided for those staining.We have supplmented the actual statistical values in the annotations as mean±SEM of each staining.

9) The authors used cultured VSMCs within passages 5-15 for their experiments. Considering the high number of passages of SMCs after the initial seeding, did the author check those cells for alpha-actin SMC to ensure that the cultured cells still show the SMC phenotype?

Response:In this study, we did overlook the issue of SMC phenotype and did not detect the expression of α- SMA.So, we resuscitated the 5th and 15th generation of smooth muscle cells, and detected the expression of α-SMA by immunofluorescence. On the one hand, the morphological changes can be observed by immunofluorescence detection, on the other hand, the expression of α-SMA can be detected. Statistical analysis showed that there was no significant difference in the expression of α-SMA between the 5th generation and the 15th generation of smooth muscle cells, suggesting that the 15th generation of smooth muscle cells still had SMC phenotype.The results are as follows:

10) Figure 2: It is recommended to provide a magnified image of positively stained areas and use arrows to mark them.

Response:According to the reviewer’s recommendation,we have provided a magnified image of positively stained areas and used arrows to mark them in Figure 2.

11) Figure 3A & 3B: aortic tissue cross sections presented in figure 3A and 3B do not correlate. Are these sections prepared from the same aortic sections?

Response:The sections of figure 3A and figure 3B are not from the same aortic sections,but the same group.

12) In figure 3, the authors only provide images of IHC for markers of pyroptosis. However, they have not performed any image and statistical analysis to quantify the expression levels for those markers. Despite the lack of analysis, at the end of sub-section 3.4, the authors concluded that “CTSB aggravated pyroptosis of VSMCs in APOE-/- mice plaques”. I don’t believe that the provided evidence warrants such conclusions.

Response:We are very sorry for our negligence of the statistical analysis lackage in figure3.We have supplemented the corresponding statistical analysis in Figure 3.

13) I couldn’t find any reference in the results section to data presented in Figure 5.

Response:It is our negligence of no mention of Fig5 in results section,we are deeply sorry about this.In order to ensure the logic of the text, we rewrite the results 3.3 to 3.5, we adjust the order of figure, and check the reference to the figure lend in the results section.

14) The authors claimed that SN50 treatment leads to decrease phosphorylation of IKB. To my knowledge, SN50 blocks the translocation and therefore activation of NFKBp65 into the nucleus. What is the relationship between SN50 mechanism of action and the pathway controlling IKB phosphorylation in the cytoplasm?

Response:To the best of our knowledge, SN50, as a synthetic protein polypeptide, prevents NFκB nuclear translocation by competitively binding to the NFκBp50 subunit in a concentration-dependent manner, and SN50 does not affect the degradation of IKB.Unfortunately, there is no report on the direct mechanism of SN50 affecting IKB phosphorylation.However, we know that the classical IKB phosphorylation process depends on IKK, and many inflammatory factors, including pyroptosis pathway factors such as IL-18[1], IL-1β[2] and TNF-α[3], have been shown to promote IKB phosphorylation activation through IKK.In short, inflammatory factors / IKK / IKB / NFκB can form a positive feedback closed loop. On the one hand, inflammatory factors promote NFκB activation, on the other hand, activated NFκB promotes the production of inflammatory factors.We can only speculate that SN50 blocks the positive feedback loop formed by inflammatory factors / IKK / IKB / NFκB by inhibiting NFκB nuclear translocation and activation, so we observed that SN50 affects IKB phosphorylation.Wang et.al[4] found that SN50 reduced TNF-α-mediated p-p65, IKKα / β, p-IKB-α elevation by overexpressing TNF-α in human umbilical vein endothelial cells and using the NFκB inhibitor SN50. This study observed that SN50 reduced p-p65 and p-IKB-α consistent with this study.

Reference for the response are as following:

1.Valente AJ, Yoshida T, Izadpanah R, Delafontaine P, Siebenlist U, Chandrasekar B. Interleukin-18 enhances IL-18R/Nox1 binding, and mediates TRAF3IP2-dependent smooth muscle cell migration. Inhibition by simvastatin. Cell Signal. 2013 Jun;25(6):1447-56. doi: 10.1016/j.cellsig.2013.03.007. Epub 2013 Mar 27. PMID: 23541442; PMCID: PMC3714795.

2.Cai J, Guan H, Jiao X, Yang J, Chen X, Zhang H, Zheng Y, Zhu Y, Liu Q, Zhang Z. NLRP3 inflammasome mediated pyroptosis is involved in cadmium exposure-induced neuroinflammation through the IL-1β/IkB-α-NF-κB-NLRP3 feedback loop in swine. Toxicology. 2021 Apr 15;453:152720. doi: 10.1016/j.tox.2021.152720. Epub 2021 Feb 13. PMID: 33592257.

3.Long J, Sun Y, Liu S, Yang S, Chen C, Zhang Z, Chu S, Yang Y, Pei G, Lin M, Yan Q, Yao J, Lin Y, Yi F, Meng L, Tan Y, Ai Q, Chen N. Targeting pyroptosis as a preventive and therapeutic approach for stroke. Cell Death Discov. 2023 May 10;9(1):155. doi: 10.1038/s41420-023-01440-y. PMID: 37165005; PMCID: PMC10172388.

4.Wang N, Jiang D, Zhou C, Han X. Alpha-solanine inhibits endothelial inflammation via nuclear factor kappa B signaling pathway. Adv Clin Exp Med. 2023 Feb 8. doi: 10.17219/acem/158781. Epub ahead of print. PMID: 36753375.

Minor Comments:

1) The manuscript needs some re-writing in proper scientific English language. Throughout the manuscripts there are so many errors with respect the proper English words or phrase and spacing in the beginning of sentences or before the brackets.

Response:We apologize for the trouble caused by language and format problems.The manuscript had be reviewed and edited by language services of Editage which recommended by journal.

2) When referring to other authors throughout the manuscript use the proper reference (e.g., Li et. al)

Response:We have made correction according to the Reviewer’s comments.

3) In the text, all “in vitro” or “in vivo” phrases should be in Italics.

Response:We have made correction according to the Reviewer’s comments.

4) Provide the full name of all markers and compounds for the first time in your text. Some of the abbreviated words are not known to all readers.

Response:Thanks for the kind reminds,we have made correction according to the Reviewer’s comments.

5) Throughout the manuscript there are many phrases that need to be corrected and revised. For example, “antigen repair” should be replaced with “antigen retrieval”, “cell-attached slides” with “tissue cross sections”, etc….

Response:Thanks for the kind reminds,we have made correction according to the Reviewer’s comments.

6) There seem to be redundancies in subsection 3.3 & 3.4 under the results section. I suggest the authors combine these two subsections.

Response:We have re-written this part according to the Reviewer’s suggestion.

7) Figure 3: on the first line of legend, “plaue pyropsis” should be corrected to “plaque pyroptosis”.

Response:Thanks for the kind reminds,we have made correction according to the Reviewer’s comments.

---

## [Decision Letter · Decision Letter 1]

8 Sep 2023

PONE-D-23-17730R1Cathepsin B aggravates atherosclerosis in ApoE-deficient mice by modulating

vascular smooth muscle cell pyroptosis through NF-κB / NLRP3 signaling pathwayPLOS ONE

Dear Dr. Liu,

Thank you for submitting your manuscript to PLOS ONE. After careful consideration, we feel that it has merit but does not fully meet PLOS ONE’s publication criteria as it currently stands. Therefore, we invite you to submit a revised version of the manuscript that addresses the points raised during the review process.

Please submit your revised manuscript by Oct 23 2023 11:59PM If you will need more time than this to complete your revisions, please reply to this message or contact the journal office at plosone@plos.org. Please include the following items when submitting your revised manuscript:A rebuttal letter that responds to each point raised by the academic editor and reviewer(s). You should upload this letter as a separate file labeled 'Response to Reviewers'.A marked-up copy of your manuscript that highlights changes made to the original version. You should upload this as a separate file labeled 'Revised Manuscript with Track Changes'.An unmarked version of your revised paper without tracked changes. You should upload this as a separate file labeled 'Manuscript'.If applicable, we recommend that you deposit your laboratory protocols in protocols.io to enhance the reproducibility of your results. Protocols.io assigns your protocol its own identifier (DOI) so that it can be cited independently in the future. For instructions see: https://journals.plos.org/plosone/s/submission-guidelines#loc-laboratory-protocols. Additionally, PLOS ONE offers an option for publishing peer-reviewed Lab Protocol articles, which describe protocols hosted on protocols.io. Read more information on sharing protocols at https://plos.org/protocols?utm_medium=editorial-email&utm_source=authorletters&utm_campaign=protocols.

We look forward to receiving your revised manuscript.

Kind regards,

Lingshan Gou, Ph.D.

Academic Editor

PLOS ONE

Journal Requirements:

Reviewers' comments:

Reviewer's Responses to Questions

**Comments to the Author**

1. If the authors have adequately addressed your comments raised in a previous round of review and you feel that this manuscript is now acceptable for publication, you may indicate that here to bypass the “Comments to the Author” section, enter your conflict of interest statement in the “Confidential to Editor” section, and submit your "Accept" recommendation.

Reviewer #1: All comments have been addressed

Reviewer #2: (No Response)

2. Is the manuscript technically sound, and do the data support the conclusions?

Reviewer #1: (No Response)

Reviewer #2: Yes

3. Has the statistical analysis been performed appropriately and rigorously? 

Reviewer #1: (No Response)

Reviewer #2: Yes

4. Have the authors made all data underlying the findings in their manuscript fully available?

Reviewer #1: (No Response)

Reviewer #2: Yes

5. Is the manuscript presented in an intelligible fashion and written in standard English?

Reviewer #1: (No Response)

Reviewer #2: Yes

6. Review Comments to the Author

Reviewer #1: (No Response)

Reviewer #2: The manuscript is improved and many of my concerns have been addressed. There are a few areas that I still have concerns:

1) the authors still have not addressed the logic for not including the female group in their study. In their response to my major comment #3, they also further confirmed the fact that there are indeed differences between male and female mice in the extent of aortic plaque formation. But, they do not address their reasons for not including female groups in their study.

2) The authors mentioned that they have included picture of CTSB protein expression in mouse aorta at 20 weeks of age in figure 2C, but I was not able to locate those figures.

3) I could not locate the alpha-actin SMC staining of the cultured SMCs that the author mentioned in their response. Also, in their response to my comment #9, they refer to their results at the end, but I couldn't locate those either.

7. PLOS authors have the option to publish the peer review history of their article (what does this mean?). If published, this will include your full peer review and any attached files.

Reviewer #1: No

Reviewer #2: No

---

## [Author Response · Author response to Decision Letter 1]

12 Oct 2023

Response to Academic Editor :

Journal Requirements:

Response: We have checked the list of references and have verified that it is correct and complete. Papers that have been retracted are not cited. This revision does not change the reference list.

*Response to Reviewers:

Reviewer #1:(No Response)

Reviewer #2: 

Major Comments:

1) the authors still have not addressed the logic for not including the female group in their study. In their response to my major comment #3, they also further confirmed the fact that there are indeed differences between male and female mice in the extent of aortic plaque formation. But, they do not address their reasons for not including female groups in their study.

Response: We are sorry for the previous unclear description. Briefly, in the most frequently used atherosclerotic mouse models, apolipoprotein E knockout (Apoe−/−) and low-density lipoprotein receptor knockout (Ldlr−/−) mice, female mice fed normal chow or atherosclerotic diet for less than 6 months generally have a greater atherosclerotic lesion size and overall burden than their male counterparts. In our study, the Apoe−/− mice were euthanized at less than 6 months of age (20 weeks), and the female atherosclerotic plaques are expected to be larger than the male atherosclerotic plaques at this age, which may have affected our experimental results. Considering the possible impact of sex differences in experimental animals on the results and the epidemiological differences in the incidence of atherosclerosis in males, our study only included male mice.

We have thought about addressing the problem of including males and females in future experiments. One possibility is to increase the sample size and group the mice (both male and female) randomly. The random grouping method should work but will require a sufficiently large sample size. If the sample size is too small, the influence of sex differences will impact the ability to obtain statistically significant findings. The second possibility is to group the female mice separately, and then compare the differences between males and females. However, the focus of such a study will be biased towards sex comparisons rather than mechanistic research. Regardless of the chosen method, studies involving both males and females are beyond the resource limitations of the current study and are planned for future experimentation.

2) The authors mentioned that they have included picture of CTSB protein expression in mouse aorta at 20 weeks of age in figure 2C, but I was not able to locate those figures.

Response: We are very sorry for our negligence regarding the picture that we mentioned, which was not included in the manuscript and reply letter. Due to figure space limitations, we intended to put the picture in the reply letter and provide the western blot quantification of CTSB in Figure 2C. However, because of issues displaying the image in the editorial system, we placed the picture in Figure 1A but forgot to change our response to the reviewer accordingly. We are sorry for the inconvenience this has caused. The plotted results in Figure 2C show that CTSB expression persisted at 20 weeks. The results in Figure 1A show that plaques appeared in the aorta of virus-transduced mice fed high-fat diet, and EGFP fluorescence was present in the plaques. For convenience, Figure 1 is persented in the file of response to reviewers and uploaded as figure1 in this system.

3) I could not locate the alpha-actin SMC staining of the cultured SMCs that the author mentioned in their response. Also, in their response to my comment #9, they refer to their results at the end, but I couldn't locate those either.

Response: We are sorry about the picture that was missing from the revised manuscript. We believe that the problem occurred when uploading files to the editorial system. We re-examined these images and added them to Figure 1I and 1J to ensure that they can be found by the reviewer. The results are persented in the file of response to reviewers and uploaded as figure1 in this system..

We tried our best to improve the manuscript by addressing reviewer concerns. We appreciate the helpful and earnest critiques of the Editors and reviewers and hope that the revisions will meet with their approval.

We would like to thank you for allowing us to resubmit a revised copy of the manuscript and we highly appreciate your time and consideration. Please do not hesitate to contact us if there are any questions. Thank you again for your hard work! Best wishes to you!

Sincerely.

Danan Liu.

---

## [Decision Letter · Decision Letter 2]

3 Nov 2023

Cathepsin B aggravates atherosclerosis in ApoE-deficient mice by modulating vascular smooth muscle cell pyroptosis through NF-κB / NLRP3 signaling pathway

PONE-D-23-17730R2

Dear Dr. Liu,

We’re pleased to inform you that your manuscript has been judged scientifically suitable for publication and will be formally accepted for publication once it meets all outstanding technical requirements.

Kind regards,

Lingshan Gou, Ph.D.

Academic Editor

PLOS ONE

Additional Editor Comments (optional):

Reviewers' comments:

Reviewer's Responses to Questions

**Comments to the Author**

1. If the authors have adequately addressed your comments raised in a previous round of review and you feel that this manuscript is now acceptable for publication, you may indicate that here to bypass the “Comments to the Author” section, enter your conflict of interest statement in the “Confidential to Editor” section, and submit your "Accept" recommendation.

Reviewer #2: All comments have been addressed

2. Is the manuscript technically sound, and do the data support the conclusions?

Reviewer #2: Yes

3. Has the statistical analysis been performed appropriately and rigorously? 

Reviewer #2: I Don't Know

4. Have the authors made all data underlying the findings in their manuscript fully available?

Reviewer #2: Yes

5. Is the manuscript presented in an intelligible fashion and written in standard English?

Reviewer #2: Yes

6. Review Comments to the Author

Reviewer #2: The major concerns are adequately addressed. I thank the author for improving the manuscript and addressing my comments. The new version of the manuscript can provide valuable information in the field. It is recommended that the authors design a future study that would include both male and female sexes included.

7. PLOS authors have the option to publish the peer review history of their article (what does this mean?). If published, this will include your full peer review and any attached files.

Reviewer #2: No

---

## [Editor Report · Acceptance letter]

19 Dec 2023

PONE-D-23-17730R2 

PLOS ONE

Dear Dr. Liu, 

I'm pleased to inform you that your manuscript has been deemed suitable for publication in PLOS ONE. Congratulations! Your manuscript is now being handed over to our production team.

Kind regards, 

on behalf of

Dr. Lingshan Gou 

Academic Editor

PLOS ONE